# Methamphetamine Induces the Release of Proadhesive Extracellular Vesicles and Promotes Syncytia Formation: A Potential Role in HIV-1 Neuropathogenesis

**DOI:** 10.3390/v14030550

**Published:** 2022-03-07

**Authors:** Subhash Chand, Catherine DeMarino, Austin Gowen, Maria Cowen, Sarah Al-Sharif, Fatah Kashanchi, Sowmya V. Yelamanchili

**Affiliations:** 1Department of Anesthesiology, University of Nebraska Medical Center, Omaha, NE 68198, USA; subhash.chand@unmc.edu (S.C.); austin.gowen@unmc.edu (A.G.); 2Laboratory of Molecular Virology, School of Systems Biology, George Mason University, Manassas, VA 20110, USA; cdemarin@gmu.edu (C.D.); mcowen4@gmu.edu (M.C.); salshar@gmu.edu (S.A.-S.); 3Department of Clinical Laboratory Sciences, College of Applied Medical Sciences, King Saud Bin Abdulaziz University for Health Sciences, Jeddah 21423, Saudi Arabia

**Keywords:** extracellular vesicles, HIV, methamphetamine, CNS, syncytia, macrophage, antiretroviral therapy, HAND

## Abstract

Despite the success of combinational antiretroviral therapy (cART), the high pervasiveness of human immunodeficiency virus-1 (HIV)-associated neurocognitive disorders (HAND) poses a significant challenge for society. Methamphetamine (meth) and related amphetamine compounds, which are potent psychostimulants, are among the most commonly used illicit drugs. Intriguingly, HIV-infected individuals who are meth users have a comparatively higher rate of neuropsychological impairment and exhibit a higher viral load in the brain than infected individuals who do not abuse meth. Effectively, all cell types secrete nano-sized lipid membrane vesicles, referred to as extracellular vesicles (EVs) that can function as intercellular communication to modulate the physiology and pathology of the cells. This study shows that meth treatments on chronically HIV-infected promonocytic U1 cells induce the release of EVs that promote cellular clustering and syncytia formation, a phenomenon that facilitates HIV pathogenesis. Our analysis also revealed that meth exposure increased intercellular adhesion molecule-1 (ICAM-1) and HIV-Nef protein expression in both large (10 K) and small (100 K) EVs. Further, when meth EVs are applied to uninfected naïve monocyte-derived macrophages (MDMs), we saw a significant increase in cell clustering and syncytia formation. Furthermore, treatment of MDMs with antibodies against ICAM-1 and its receptor, lymphocyte function-associated antigen 1 (LFA1), substantially blocked syncytia formation, and consequently reduced the number of multinucleated cells. In summary, our findings reveal that meth exacerbates HIV pathogenesis in the brain through release of proadhesive EVs, promoting syncytia formation and thereby aiding in the progression of HIV infection in uninfected cells.

## 1. Introduction

Human immunodeficiency virus-1 (HIV)-associated neurocognitive disorders (HAND) are found in nearly one-third of HIV^+^ patients and pose a significant challenge for society, even with the availability of combinational antiretroviral therapy (cART) [1,2,3,4]. Methamphetamine (meth), mephedrone, ketamine, and gamma-hydroxybutyrate are the most common recreational drugs associated with chemsex and pose a significant challenge in HIV prevention and treatment [5,6,7]. These drugs are used to boost, prolong, and intensify sexual experiences [5]. However, in sexual settings, these drugs are linked to the acquisition and transmission of HIV and other sexually transmitted diseases (STDs) [8,9,10]. Meth is a highly addictive central nervous system (CNS) stimulant that induces structural and biochemical alterations to the brain [11]. With increasing meth users worldwide, it poses a threat to overall health as well as the economy. The National Institute on Drug Abuse (NIDA) reports that overdose death rates involving meth have quadrupled from 2011 to 2017 (https://www.samhsa.gov/find-help/atod, accessed on 17 February 2022). During 2015–2018, about 1.6 million adults aged 18 or above have reported the use of meth in the United States [12]. Meth enhances HIV replication, antiviral drug resistance, and progression to AIDS. Compared to non-drug users, individuals with a drug use history significantly have a higher viral load [13]. In addition, HIV-positive meth users are less likely to adhere to cART and are at high risk of viral load rebound [14,15,16,17]. Further, in monkeys infected with simian immunodeficiency virus (SIV), it has been shown that meth treatment significantly increases brain viral load [18]. However, the impact of meth on HIV infection and disease progression in CNS are yet to be fully elucidated. 

In recent times, extracellular vesicles (EVs) have gained considerable attention as novel actors in intercellular communication and disease dissemination [19,20,21,22,23]. EVs have been implicated in HIV pathogenesis, neuroinflammation, and HAND. The presence of viral components in EVs secreted from infected cells signifies their potential role in mediating infection. For instance, EVs secreted from HIV infected cells have gp120 envelope (Env) protein that initiates virus attachment and fusion to the host cells [24,25], thus facilitating HIV infection [26]. Further, ex vivo depletion of Env containing EVs from the infection prep decreases HIV infection in human lymphoid tissue [26]. EVs have also been shown to activate the transcription of HIV in the latently infected cells and promote viral infection [27,28,29]. Several studies have also noted the role of EVs in inflammation [30,31], immune activation [31,32], HIV transmission [30,33], and maintenance of the HIV reservoir in the CNS [31].

Interestingly, multinucleated cells (MNC) and syncytia formation have been observed during HIV and simian immunodeficiency virus (SIV) infections [34,35,36]. Studies from Najera et al. have reported that meth and SIV infection have an additive effect on the development of multinucleated giant cells in the brain [37]. Although it was clear that syncytia and MNC formation led to transmission of virus to uninfected cells in the brain, no studies so far have described the exact mechanism/s promoting the formation of these structures in the brain. Here, we aimed to identify mechanisms by which meth-induced EVs play a role in the progression of HIV infection. Specifically, we focus on deciphering cellular receptors that might be involved in cell clustering and syncytia formation. Our data revealed that the meth-treated MDMs secreted a considerably higher number of EVs, further enhancing cell clustering and syncytia formation in naïve uninfected MDMs. Our western blot analysis showed increased expression of intercellular adhesion molecule 1 (ICAM-1), vascular cell adhesion molecule (VCAM), and Nef protein in meth and HIV infected MDMs. Further, treatment of EVs with anti-lymphocyte function-associated antigen 1 (LFA1) or anti-ICAM-1 antibody (Ab) significantly decreased the formation of multinucleated cells, indicating their possible role in cell clustering and syncytia formation.

## 2. Materials and Methods 

### 2.1. Cells

Human promonocytic (U937) and chronically infected HIV-1 promonocytic (U1) cell lines were cultured in RPMI-1640 media (Quality Biological) supplemented with 10% fetal bovine serum (FBS), 2 mM L-glutamine, 100 U/mL penicillin, and 100 mg/mL streptomycin at 37 °C with 5% of CO_2_. Once the cell count reached around a hundred million, cells were centrifuged at 1200 rpm for 5 min and were resuspended in 100 mL extracellular free RPMI media. To differentiate them into macrophages, these cells were treated with 100 nM phorbol 12-myristate 13-acetate (PMA) (LC Laboratories; P-1680) every other day for 7 days. The media was exchanged every other day. After day seven, cells were grown in media devoid of PMA and were treated with meth, meth + cART, or no treatment as per their respective group. On day 13, supernatant and cells were collected. The 10 K and 100 K EVs were isolated from supernatant following a previously published protocol [38]. 

Primary monocytes were obtained from the Department of Pharmacology and Experimental Neuroscience, University of Nebraska Medical Center (UNMC), Omaha, NE, USA. These cells were grown in DMEM (Dulbecco’s Modified Eagle’s Medium # 10-013-CV) media containing 0.1 ng/mL macrophage colony-stimulating factor (M-CSF). The culture media was half exchanged every alternate day. M-CSF was removed from the growing media when cells were fully differentiated to macrophages on day seven. These monocyte-derived macrophages (MDMs) were used either for syncytia experiments or were infected with HIV/meth/cART for the different experimental groups (control, 10 µM meth, 50 µM meth, 10 µM meth + cART, 50 µM meth + cART). 

### 2.2. Meth and cART Treatment

For drug treatment groups, differentiated cells were treated with meth (10–250 µM) for 48 h (two doses at 24 h intermission that replicates a in vivo “binge” scenario) in EV-free RPMI media. For meth + cART groups, cells were treated with two doses of meth at 24 h intervals, followed by two doses of cART treatment at an interval of 48 h. The cART cocktail contains 10 nM concentration of Tenofovir (NIH AIDS Reagent Program catalog # 10199), Emtricitabine (AIDS Reagent Program catalog # 10071), Darunavir (Prezista, TMC 114) (AIDS Reagent Program catalog # 11447) and 5 nM concentration Ritonavir (AIDS Reagent Program catalog # 4622). For all groups, the cells and supernatants were collected on the 13th day. The cell culture supernatants were used for EV isolation and further analysis. 

### 2.3. Cell Fixing and Staining

Cells on coverslips were fixed using 4% paraformaldehyde (PFA) (Acros Organic CAS30525-89-4) for 15 min. After fixation, cells were permeabilized using 0.25% Tween 20 (Fisher BP337-500) for 15 min. Cells were then blocked for 1 h at room temperature with blocking solution (5% of normal goat serum (NGS), 1% BSA, and 0.25% Tween 20 in PBS). Coverslips were washed three times for 5 min each with PBS. Cells were then incubated with Abs Phalloidin Alexa Fluor 568 (# A10037; 1:50 dilution), KC57 FITC (Beckman Coulter # CO660466; 1:30 dilution), and 4′,6-diamidino-2-phenylindole (DAPI) (0.3 µM) for 30 min in the dark. Finally, coverslips were washed three times with PBS and one time with DI water. Subsequently, coverslips were mounted with the prolonged antifade (Invitrogen P36930) and kept in the dark before imaging.

### 2.4. Extracellular Vesicles Isolations

U1 and U937 cells were grown in RPMI media supplemented with 10% EV-free FBS, and EVs were isolated from 100 mL of cell culture supernatant. Cells were pelleted by centrifugation at 300× *g* for 10 min to collect the cell supernatant. The supernatant was collected, and ultracentrifugation in a Ti70 rotor (Beckman Coulter; Indianapolis, IN, USA) was performed at 10,000× *g* for 45 min and 100,000× *g* for 90 min, to pellet 10 K and 100 K EVs, respectively, at 4 °C. All pellets were then resuspended in a particle-free PBS (Dulbecco’s phosphate-buffered saline without calcium and magnesium) and washed with PBS. The resulting pellet was resuspended in 300 µL of PBS. For EV and virion isolation from samples, we utilized Nanotrap particles as published previously [39,40]. Briefly, equal amounts of Nanotrap particle (NT80), which precipitates the only EVs not HIV and PBS (1X without calcium and magnesium) were mixed and resuspended to make a slurry. To capture EVs and virions from supernatants, 60 µL slurry was added to 1 mL supernatant and rotated overnight at 4 °C. The particles were separated, washed with PBS, and the pellets were resuspended in 50 µL PBS and used for downstream assays.

### 2.5. Nanoparticle Tracking Analysis (NTA)

EV size distribution and concentration measurements were done as published previously [41,42]. Briefly, 10 μL of the EV sample was diluted to 1:100–1:1000 in PBS and were injected into the instrument. The instrument was equipped with a syringe pump and a 488 nm laser. The measurement option was selected as per the manufacturer’s advice for the capture of the videos. Particle-free PBS was used for background measurements. Five videos were recorded for each EV preparation, and NTA 3.1 version software was used for analysis.

### 2.6. Western Blot

The protein samples were loaded onto a 4–20% Tris-glycine gel (Invitrogen; Pittsburg, PA, USA) and were run for 30 min at 180 V. Overnight protein transfer onto Immobilon membranes (Millipore; Burlington, MA, USA) was performed at 50 mA. Subsequently, membranes were blocked for 30 min with PBS-T (PBS containing 0.1% Tween 20 and 5% dry milk) at 4 °C. Membranes were incubated with appropriate primary antibody (Ab) overnight at 4 °C on a rocker. Next day, membranes were washed thrice with PBS-T and incubated with appropriate HRP-conjugated secondary Ab in PBS-T for 2 h at room temperature. After incubation, membranes were washed twice with PBS-T, once with PBS, developed with Clarity Western ECL Substrate (Bio-Rad; Hercules, CA, USA), and visualized by the Molecular Imager ChemiDoc Touch system (Bio-Rad).

### 2.7. Labeling of Extracellular Vesicles and Confocal Microscopy

EVs were purified from U1 and U937 cell culture and separated into 10 K and 100 K EVs. A 1.5 μL fluorescent label of BODIPY™ 493/503 (Cat. # D3922; Invitrogen™) was mixed with 50 μL EVs and incubated for 30 min at 37 °C. Any unbound BODIPY was filtered out using a Pharmacia G-50 spin column (1 mL bed volume in PBS buffer; 2000 rpm/2 min; Sorval RT6000D), yielding 30 μL of labeled EVs. In biological triplicate, five microliters of labeled EVs were added onto MDMs (50,000 cells on each coverslip in 200 μL at cell: EV ratios 1:10,000). Treated cells were analyzed with confocal microscopy at the UNMC core facility. The prolong gold antifade mounted slides were imaged in Zeiss Observer.Z1 microscope equipped with a monochromatic Axiocam MRm camera using Axiovision 40 v.4.8.0.0 software (Carl Zeiss, Oberkochen, Germany). The red, green, and blue colors were assigned to Alexa Fluor 568, KC57-FITC, and DAPI, respectively.

### 2.8. Cell Contact Inhibition Assay

Cell contact inhibition was performed using 20 µg/mL of ICAM-1 Ab (Santa Cruz Biotechnology, sc-8439) and 10 µg/mL LFA-1 Ab (Invitrogen, 16-0119-81) as described previously [43]. The ICAM-1 Ab was applied on EVs and incubated at 4 °C on a rocker overnight, while LFA-1 Ab was applied to MDMs and was incubated overnight at 37 °C. After Ab treatments, respective EV groups were applied to MDMs and were incubated for 3 h before fixing and staining to visualize the blocking of cell clustering and syncytia formation.

### 2.9. Cell Toxicity Assay

For the cell toxicity assay, 100 µM meth was added to the U937 cell culture for 24 h, while the control cell culture did not receive any meth. After 24 h, LDH (lactate dehydrogenase) assay was performed on the media using the kit (Cytotoxicity detection kit (LDH), Roche, Basel, Switzerland), as described previously [44]. In brief, 100 μL of cell culture medium was transferred to a 96-well plate to which 100 μL of the reaction solution containing catalyst and detection dye were added from the kit. After 30 min of incubation, absorption was measured at 490 nm with 655 nm as a reference wavelength. Two percent triton was used for positive control, and the cytotoxicity was calculated following the equation: $Cytotoxicity\left(\%\right)=\frac{\left(Experimental\;value-Media\;control\right)}{\left(Positive\;control-Media\;control\right)}*100$.

### 2.10. RNA Isolation and qPCR

RNA was isolated using Trizol (Life Technologies, 15596026) according to the manufacturer’s instructions. According to the manufacturer’s instructions, TaqMan miRNA and gene expression assays were used for cDNA synthesis and real-time PCR (RT-PCR) (Life Technologies, Carlsbad, CA, USA). Small nuclear RNA U6 (U6) and glyceraldehyde 3-phosphate dehydrogenase (GAPDH) were used as a control for mRNA studies. Delta-delta (ΔΔ) Ct method was used to calculate fold change, as described previously [41,44,45].

### 2.11. Syncytia Count

All images for counting purposes were taken using a 10x objective under an EVOSM500 microscope using DAPI and differential interference contrast (DIC). The capture was taken at identical points of the coverslip when capturing images. All slides contained five images to account for potential image issues and variability. The software ImageJ was used for the counting process (settings: image type—8 Bit, image threshold baseline—150, analyze particles 50–250 to count normal cells and 251–2000 to count multi nuclei cells). Nuclei size was determined by visual analysis of DAPI staining in control images for both cell groups. Fifty particles were set as a minimum to eliminate any potential background staining from counting towards the syncytia counts. The average nuclei size of a non-multi nuclei cell was determined to fall in a range of 140–175 pixels. Allowing us to choose 251 as a baseline for a multi-nuclei body, and 2000 was used as a maximum because no nuclei group would extend past that pixel count. The calculation accounted for the number of nuclei in each range and the size of each DAPI pixel range, giving an average size for single nuclei and multi-nuclei bodies. All size-based calculations were performed by taking the average size of the nuclei in the non-syncytia group for each capture. To calculate the number of nuclei in a multi-nuclei body, the average size of multi-nuclei bodies in the photograph was divided by the average size of single nuclei in the same picture to estimate the average number of nuclei in the multi-nuclei groups.

### 2.12. Statistics

The statistical analyses were done using GraphPad. *p*-values were calculated using a two-way ANOVA and were significant when *p* values were *p* < 0.05 (*), *p* < 0.01 (**), and *p* < 0.001 (***). Multiple *t*-tests were performed followed by post hoc analysis as required.

## 3. Results

### 3.1. Impact of Meth on EVs Biogenesis in HIV Infected Macrophages

It has been well documented that meth exacerbates HIV infection; however, the mechanisms are poorly understood [46,47,48]. A previous study showed that HIV infection increases EV release in microglia [49]. We have previously demonstrated that meth increases EV releases in animal (rat and monkey brain tissues) and cell culture models [50]. However, the effects of meth on EV release in HIV infected cells have not been investigated. To understand the effects of meth on HIV infection, we utilized uninfected promonocytic cell line U937 and chronically HIV infected promonocytic cell line U1. U1 cell line is used as a model for provirus latency in myeloid cells, and the virus can be reactivated upon cell stimulation with TNF-α/-β or PMA [51]. Since brain is now considered as viral reservoir [52], we chose to use this model to recapitulate chronic HIV infection in brain. We first tested whether meth treatments at different concentrations affected the cellular toxicity of U937 cells. Lactate dehydrogenase assay (LDH) was performed on media supernatants of meth treated U937 cells. Results indicated that only at higher concentrations (500 μM) meth was toxic to uninfected cells (Figure 1A). Next, we isolated EVs from U937 cells treated with meth concentrations varying from 10–250 μM. In order to investigate the EV subtype release dynamics after meth treatments, we isolated both the large EVs (10 K) and small EVs (100 K) from cells treated with two doses of meth for 48 h in (which replicates a in vivo “binge” scenario). Nano tracking particle analysis (NTA) revealed that there was no difference in 10 K EV secretion (Figure 1B); however, a significant difference was seen with 100 K EVs in all meth treatments (10–250 μM). treated cells released significantly more particles than the untreated controls (Figure 1C). Next, similar experiments were performed in U1 cells. LDH assay revealed that U1 cells were significantly more sensitive to meth treatments. At higher concentrations (100–500 μM), meth caused significant toxicity (Figure 1D). Finally, we used NTA to quantify the particle concentrations in both 10 K and 100 K EV pellets. Interestingly, we see a significant increase in both 10 K and 100 K EV when treated with 10 and 50 μM meth concentrations over 48 h (Figure 1E,F).

### 3.2. Meth Alters Genes in ESCRT-Dependent and Independent Pathways

Due to the observed increase in EV secretion after meth treatments in both U937 and U1 cells, next, we investigated the effects of meth on gene expression of the proteins central to EV biogenesis. Several pathways are involved in EV biogenesis [53,54,55,56,57,58] which are primarily categorized into the endosomal sorting complex required for transport (ESCRT)-dependent pathway [59] and ESCRT-independent pathway involving ceramide synthesis [60]. We aimed to screen the genes involved in both pathways of EV biogenesis. Since half-life of meth is ~10–11 h [61], we chose to treat the cells for 24 h with one dose of 100 μM meth to identify genes that are activated immediately after acute drug exposure. Total RNA was isolated and used to synthesize a complementary DNA (cDNA) library, followed by qPCR, to observe gene expression changes compared to the respective untreated control groups. Our results showed a significant increase in the genes for the ESCRT-dependent pathway, specifically those belonging to ESCRT-0 (HGS, STAM2), ESCRT-I (VPS37B), and the disassembly complex (VPS4A) in U937 cells. Genes involved in ESCRT-independent EV biogenesis, CERS1, CERS2, CERS3, CERS5, and CERS6, were also significantly upregulated in meth-treated U937 cells as compared to control (Figure 2A). Proinflammatory cytokines, IL1β, and TNFα were upregulated considerably in meth-treated U937 cells (Figure 2A). A similar experiment on HIV-1 infected U1 cells revealed that genes involved in ESCRT-dependent (ESCRT-0: STAM, STAM2; ESCRT-I: VPS37A and VPS37B; ESCRT-II: VPS36 and ESCRTIII: CHMP3), disassembly complex (VPS4B, Alix), and ESCRT-independent (CERS1, CERS2, and CERS6) and proinflammatory cytokines, IL1β and TNFα were significantly upregulated in meth-treated U1 cells (Figure 2B). These significantly upregulated proteins in meth-treated groups are crucial for EV biogenesis and its secretion [62]. These data indicate that meth increases EV biogenesis in both uninfected, as well as HIV-1 infected macrophages.

### 3.3. Role of Meth on EVs Cargo and Expression of Viral Protein

There is strong evidence that meth increases the viral load in macrophages and contributes to exacerbating HAND [18]. HIV-1 proteins are released in EVs [63,64,65]; however, whether EVs released by chronic meth exposure from latent HIV infected cells exacerbates pathogenesis is unclear. To see if there is a relationship between meth, EV release, and HIV infection, we tested whether meth had any impact on the secretion of viral proteins in EVs. U937 or U1 cells were treated with two doses of meth for 48 h at 24 h intervals. Our results indicate that meth did not induce HIV protein, gp120 secretion in the 10 K or 100 K EVs isolated from U1 cells. Similarly, in U1 cells, meth did not increase the secretion of HIV accessory protein, Nef, in 10 K EVs (Figure 3A; lane 4–6); however, it induced a dose-dependent increase in Nef expression in 100 K EVs (Figure 3B; lane 4–6). Interestingly, when combined with cART, meth increased Nef secretion in 10 K EVs (Figure 3A; lane 7–8), whereas not much difference was seen in 100 K EVs (Figure 3B; lane 7–8, Appendix A). These data indicate that the presence of meth could cause an increase in Nef secretion in both large (10 K) and small (100 K) EVs and that cART could enhance the secretion of Nef in 10 K EVs. Further, we tested if meth exposure increases the release of adhesion molecules ICAM-1 and VCAM in EVs. Results showed a dose-dependent increase in ICAM-1 expression in 10 K and 100 K EVs isolated from U937 and U1 media supernatants (Figure 3A,B, Appendix A). ICAM-1, also expressed on macrophages, is the surface molecule that mediates adhesion and migration [66]. We did not observe VCAM secretion in EVs. Further, EV marker HSC 70 showed a similar increase with meth in both U937 and U1 cells, indicating an enhanced secretion of EVs in meth-treated macrophages.

### 3.4. Role of Meth and EVs in Syncytia and HIV Pathogenesis

Syncytia is a form of cell fusion where infected cells can fuse with neighboring cells to form giant multinucleated cells and facilitate the spread of the virus [67,68,69,70]. We observed that meth increased ICAM-1 (adhesion molecule, which influences cell–cell tethering) and Nef (membrane anchor) proteins in EVs. Therefore, we wanted to see whether EVs isolated from meth-treated U1 cells promote cell clustering and syncytia formation in MDMs. To test this, we isolated 10 K and 100 K EVs from U1 cells treated with meth. EVs were labeled with a green fluorescent dye, BODIPY, applied on uninfected naïve MDMs and incubated for 3 h. MDMs were then fixed and immunostained with phalloidin, an F-actin stain, to look at the overall phenotype of the cells after treatments. We observed that both 10 K and 100 K EVs from 10 and 50 µM meth-treated U1 cells promoted cell clustering and the formation of multinucleated cells or syncytia (Figure 4A). Next, we wanted to evaluate cART treatments on syncytia formation. MDMs were treated with 10 K and 100 K EVs isolated from the meth in combination with cART treated U1 cells. Our data revealed that both 10 K and 100 K EVs from meth (10 and 50 µM) + cART-treated groups also enhanced cell clustering and syncytia formation in MDMs (Figure 4B,C). This was further confirmed by labeling the EV-treated MDMs with Kc57-FITC Ab that labels the HIV core proteins (55, 39, 33, and 24 kDa) (Figure 5). Similar to Figure 4, EVs (10 K and 100 K) from U1 cells treated with meth increased syncytia formation after 3 h (Figure 5). Further, EVs from cART treated groups also significantly enhanced syncytia formation at 3 h (Figure 5). Thus, in general, we observed EVs (10 K and 100 K) fromU937 and U1 cells-initiated syncytia formation more efficiently at 3 h (Figure 4 and Figure 5, Appendix A) and Kc57-FITC (HIV core proteins: 55, 39, 33, and 24 kDa) markers.

### 3.5. Antibody Treatment against ICAM-1 and LFA-1 Antibodies Prevented Meth-Induced Syncytia Formation

To further evaluate the mechanisms underlying syncytia formation, we investigated the interaction of ICAM-1 with its receptor, lymphocyte function-associated antigen 1 (LFA-1). To investigate this mechanism, EVs isolated from the U1 cells were treated with ICAM-1 Ab to block the interaction with its receptor (Figure 6A,B). Similarly, the recipient MDMs were treated with LFA-1 Ab (Figure 6C,D). MDMs were then fixed, stained, and imaged for syncytia counts. Results indicate that ICAM-1 treated 10 K EVs and LFA-1 Ab treated MDMs showed no change in % of syncytia in 10 K EV (10 µM and 50 µM meth) treated groups. In contrast, a significant decrease was observed in meth + cART treated 10 K EV groups (Figure 6A,C). In comparison, MDMs treated with either ICAM-1 Ab treated 100 K EVs (10 µM and 50 µM meth) or with LFA-1 Ab showed a significant decrease in syncytia blocking compared to their respective controls (Figure 6B,D). Further, a significant decrease in syncytia was also observed in the cART groups (Figure 6B,D). These results indicate that blocking ICAM-1 in 100 K EVs or LFA-1 blocking in recipient MDMs efficiently decreases syncytia formation.

## 4. Discussion

This study elucidates how HIV/meth in synergy affects EV-biogenesis and is possibly involved in disease progression in the brain. Meth is known to enhance HIV infection of macrophages and contributes to the long-term persistence of HIV in the brain [71,72]. It is also well known that meth exacerbates HIV infection; however, the mechanisms are poorly understood [46,47,48]. Nearly 15% of HIV patients in the United States acknowledge using meth [73]. Alvarez-Carbonell et al. have reported that meth can reactivate HIV in HIV-infected microglia, sensitize the microglial cells to proinflammatory cytokines, and exacerbate HAND [3]. Our current study found that meth increases the release of EVs from uninfected and chronically infected promonocytic cells lines (Figure 1). Intriguingly, in uninfected U937 cells, meth did not increase the concentration of large 10 K particles when compared to the smaller 100 K particles (Figure 1B,C). In our recent study, we showed that meth increases EV biogenesis in uninfected microglial cells, however, it increased specifically smaller sized particles when compared to larger sized particles [50]. This could be the same in uninfected U937 cells as well. Further, meth also increases the gene expression of many proteins that are central to EVs biogenesis (Figure 2). We investigated the effects of meth on EV biogenesis through analyzing the genes in the ESCRT dependent and independent pathways. ESCRT complex is consists of about thirty proteins arranged in four complexes, namely, ESCRT-0, -I, -II, and -III, along with disassembly complex proteins, such as VPS4, VTA1, and ALIX [62,74,75]. The molecular mechanisms that control the trafficking and secretion of EVs have been extensively studied [76,77,78,79,80]. ESCRT proteins (e.g., Tsg101 and HRS), lipids (e.g., ceramide), and tetraspanins (e.g., CD9 and CD81) have been demonstrated to regulate EVs secretion by regulating biogenesis and fate of multivesicular bodies (MVBs) [57,76,81]. GTPases, such as RAB11, RAB27, and RAB35, have been reported to regulate EVs, perhaps by controlling the transport and docking of MVBs to the plasma membrane [57,75,77]. Previous studies have also reported that silencing or knockdown of ESCRT genes have altered the EV composition and the numbers of EVs secreted [75]. In addition, disassembly complex proteins such as VPS4A/B, VTA1, and ALIX play a critical role in dissociation and recycling the ESCRT machinery [75]. Further, several studies have reported the role of ceramides in EV biogenesis [79,82,83]. Ceramide synthases (CERS) are the crucial regulator of de novo synthesis of long acyl chain ceramides. Astarita et al. have shown that meth stimulates the biosynthesis of ceramide and its accumulation in rat organ tissues [84]. In this study, we have seen the significant expression of CERS (CERS1, 2, 3, 5, and 6) in meth-treated groups. We have further noted that EVs numbers were substantially higher in meth-treated cells. Collectively, we found several genes essential for EV biogenesis (RAB27A, HGS, STAM1/2, ALIX, CERS1/2/3//5/6, VPS4A/B, and proinflammatory cytokines IL1β and TNFα) were significantly upregulated in meth-treated and HIV-infected cells. Thus, our finding indicates that HIV and meth enhance EV biogenesis by upregulating the ESCRT machinery-related pathway, highlighting HIV and meth synergy in EVs biogenesis and neurological disorders in PLWH.

EVs can activate microglia [85,86] and activation of microglia correlates with HAND in clinical settings [87]. Meth increases EVs biogenesis in the brain [50], and through direct and indirect mechanisms, it exacerbates the severity and onset of HAND [88]. Various studies have demonstrated that HIV proteins are released in EVs [63,64,65]. Interestingly, EV-cargo (HIV-proteins and RNA) from HIV-infected cells can be altered due to meth exposure [65]. It is widely reported that HIV does not infect neurons directly, and neurotoxicity is mediated by viral proteins, cytokines, small soluble factors released from the infected cells [89]. Sami et al. have reported that Nef-associated EVs can be taken up by primary human neurons [90]. Thus, presumably, EVs secreted from the meth-exposed infected cells promote glial cell fusions, contribute to the cell-to-cell spread of HIV, and mediate the detrimental effects of meth in worsening of HAND. Our data showed a meth dose-dependent increase in Nef expression in EVs (Figure 3). HIV-Nef can continuously modulate actin polymerization to induce morphological changes in macrophages [91]. Consequently, HIV-infected macrophages show cellular protrusions correlated with the changes in cell adhesion and migratory characteristics [92]. Interestingly, Lopez et al. demonstrated that HIV infection promotes Nef expression-dependent phenotypic changes such as enhanced podosome formation in MDMs in 3-dimensional tissue culture [91]. Microglia are CNS resident macrophages that are susceptible to HIV infection and are one of the key cellular reservoirs of latent HIV [93]. Both ICAM-1 and LFA-1 (receptor for ICAM-1) are expressed on macrophage and microglia membranes, and their interactions are indispensable for immunological synapse formation between immune cells.

Several studies on macrophages and microglia from HIV-infected individuals on cART have shown the remnants of HIV genetic materials within brain tissues [71,72,94]. These HIV reservoirs in the brain are inaccessible to circulating Abs and are not removed entirely by cART. Thus, macrophages play a critical role in HIV pathogenesis and its persistence in the CNS. Further, the persistence of HAND in individuals who are on cART suggests that HIV continues to replicate and spread in the brain [95,96]. We noted that cART failed to decrease the expression of Nef or ICAM-1 in the EVs secreted from U1 cells, these results suggest that latently infected cells are somehow resistant and could still potentially secrete HIV proteins in EVs. In addition, cART treatment was unable to reduce syncytia formation, instead, there were significant increases in multinucleated cells indicating that EVs could play a key role in exacerbating neuropathogenesis even in individuals under ART regimen. A further study investigating the role of cART on EVs cargo would be essential to understand its implications on EVs loading, syncytia formation and neuropathogenesis.

Finally, our findings on the ICAM-1 and LFA-1 interactions suggest EVs bring cells together for clustering and syncytia. Multinucleated giant cells formation, microglial activation, inflammation, loss of neurons, and astrocytes are the neuropathological characteristics of the HIV entering to the CNS [97]. Furthermore, Najera et al. have observed multinucleated giant cells (syncytia) in simian immunodeficiency virus (SIV) infected, and meth-treated macaques’ brains [37]. The enhanced release of ICAM-1 on the surface of EVs secreted from HIV-infected meth-treated cells may interact with LFA-1 of more than one macrophage and bring them together. This study and others [60] demonstrated that ICAM-1 on infected donor cells interacts with LFA-1 of the target cell (Figure 6). This interaction would be more assertive with the significantly higher presence of ICAM-1 in meth-induced EVs. Cell clustering will also depend on the number of EVs present in the extracellular space. The potential synergy between meth-induced ICAM-1 expression on EVs and secretion of a significantly higher number of EVs from the infected cells may promote cell clustering and, eventually, syncytia.

Alternatively, the LFA-1 of the target cell may be recognized by another EV that can bring another cell close to the target cell, further enhancing cell clustering. In addition, blocking EVs with ICAM-1 Ab and LFA-1 Ab significantly decreased the syncytia, indicating their role in cell clustering. In addition to this, efficient dissemination of HIV may no longer be essential to propagate neurocognitive disorders in CNS as HIV-associated cytotoxicity may well reach glial cells and neurons via EVs shuttling. Given this promiscuity of EVs, meth, and HIV, a combinatorial approach would be necessary to deal with HIV and meth-induced HAND. In addition to cART, adding LFA-1 and ICAM-I Abs may be required in the treatment regimen of HIV and HAND to mitigate EV-mediated neurotoxicity and cognitive impairments. Here, we investigated the role of 10 K and 100 K EVs in HIV neuropathogenesis. We have previously reported that EVs from the brain of meth-treated monkeys showed significant heterogeneity in size and higher concentrations than control animals [50]. However, a comprehensive study investigating the impact of different types of EVs, such as exosome, microvesicle, mitovesicle, apoptotic EVs, exomere, or low and high-density EVs might be necessary to decipher the role of EVs heterogeneity in HIV pathogenesis.

## Figures and Tables

**Figure 1 viruses-14-00550-f001:**
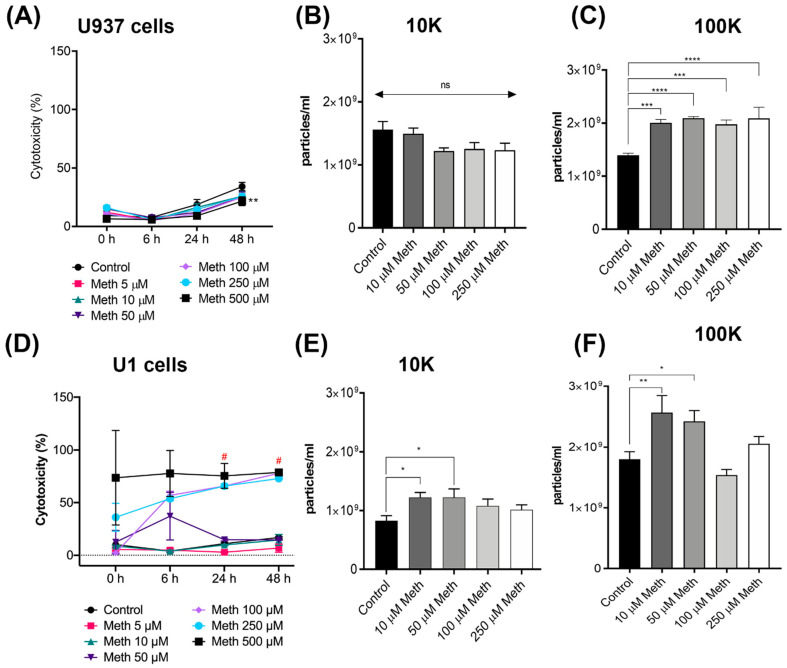
Impact of meth on cell toxicity and EV biogenesis. Cytotoxicity assay in U937 cells (**A**) treated with 0, 5, 10, 50, 100, 250, and 500 μM meth for 48 h (*n* = 3). NTA measurements for EVs concentration of 10 K-EV (**B**) and 100 K-EV (**C**). Cytotoxicity assay in U1 cells (**D**) treated with different concentrations of meth for 48 h (*n* = 3). NTA concentration measurements of U1-10 K-EV (**E**) and 100 K-EV (**F**). Data are shown as mean +/− SEM and represent at least three independent assays done in triplicate. * *p* < 0.05; ** *p* < 0.01; *** *p* < 0.001, **** *p* < 0.0001, and ns = not significant. U937: human monocyte; U1: chronically infected HIV-1 promonocytic; 10 K: 10,000× *g*; 100 K: 100,000× *g*.

**Figure 2 viruses-14-00550-f002:**
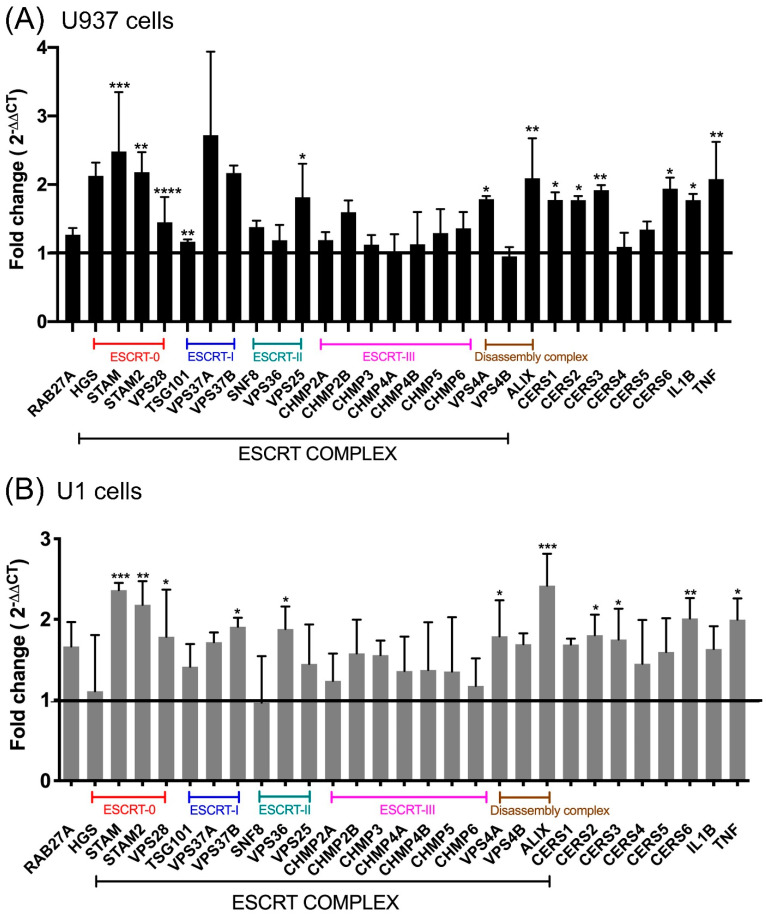
Uninfected control U937 (**A**) and chronically HIV-infected U1 cells (**B**) were treated with 100 μM meth for 24 h. qPCR was performed on the RNA isolated from the cells. Data are presented as a fold change in expression relative to untreated control. Multiple t-tests were performed to calculate statistical significance between groups *n* = 3 (biological replicates), post hoc analysis was performed using the Fisher’s LSD. * *p* < 0.05; ** *p* < 0.01; *** *p* < 0.001; **** *p* < 0.0001.

**Figure 3 viruses-14-00550-f003:**
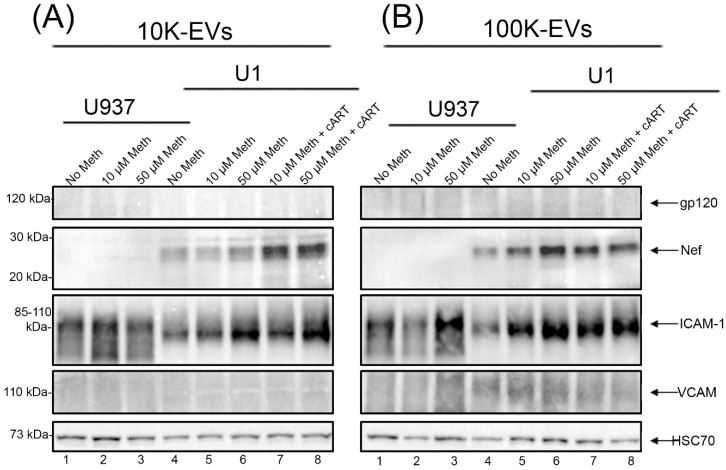
Impact of meth on EV cargo and expression of viral protein. Expression of HIV viral proteins gp120, and Nef, adhesion proteins: ICAM-1-, VCAM-, and EV-associated proteins HSC70 in 10 K (**A**) and 100 K (**B**) EVs isolated from meth-treated U937 and U1 cells culture supernatant.

**Figure 4 viruses-14-00550-f004:**
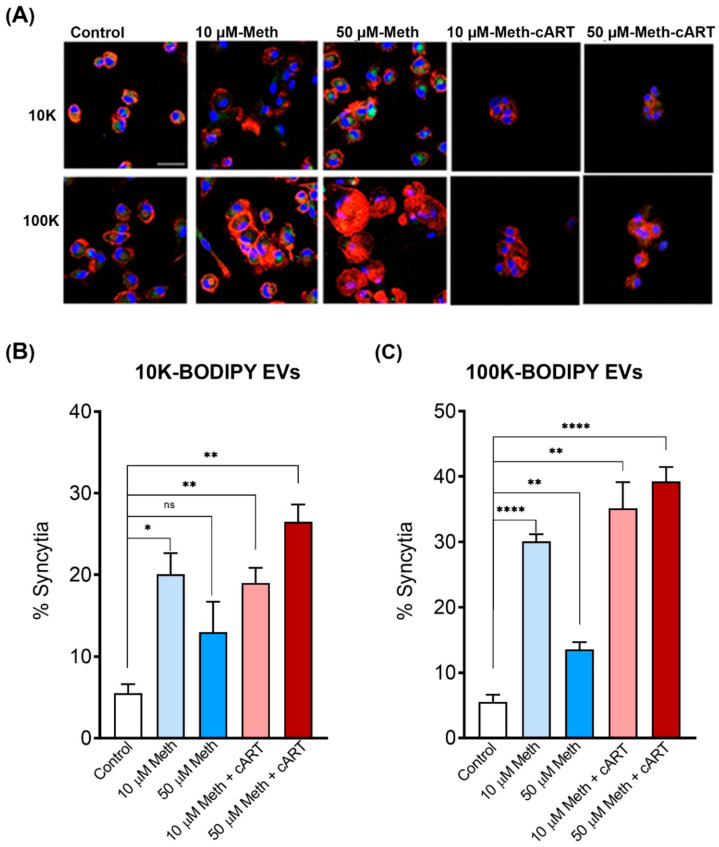
Syncytia count induced by various groups of 10 K and 100 K EV at 3 h (**A**). MDMs from 10. K (**B**) and 100 K (**C**) EV set of different U1 groups for 3 h. Data are shown as mean +/− SEM and represent at least three independent assays done in triplicate. * *p* < 0.05; ** *p* < 0.01; **** *p* < 0.0001, and ns = not significant.

**Figure 5 viruses-14-00550-f005:**
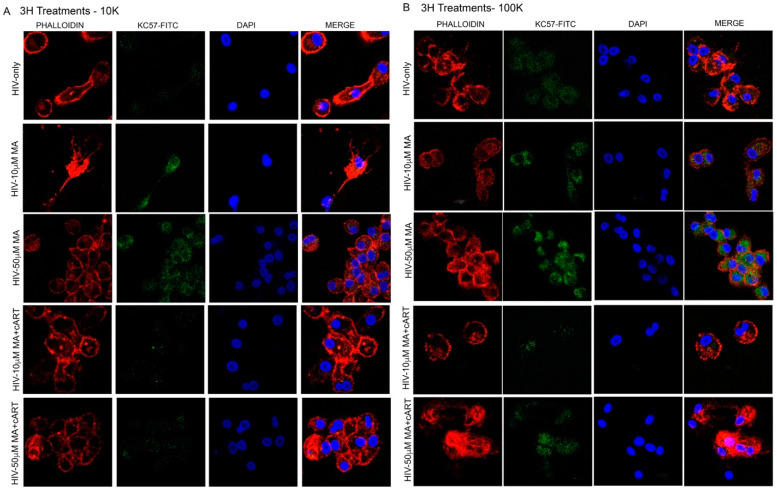
EV from HIV/meth-treated MDM promote syncytia in naïve MDMs. Syncytia formation was observed three hours post-treatment with 10K (**A**) and 100K (**B**) EV. Phalloidin (F-actin stain), Kc57-FITC (HIV core protein marker), DAPI (nucleus marker); scale bar 20 µm. N = 3 (MDMs from three independent donors).

**Figure 6 viruses-14-00550-f006:**
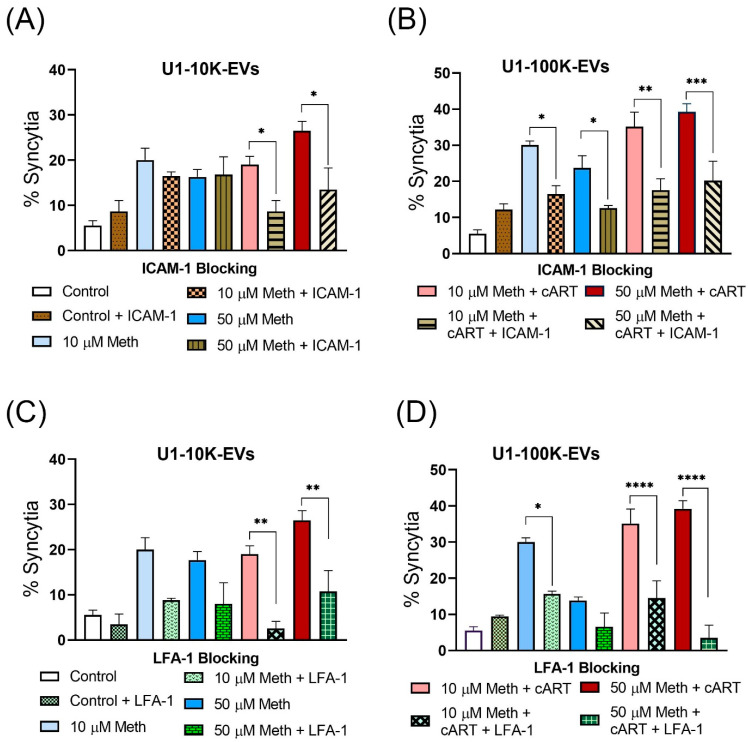
Blocking of syncytia formation by treating EVs with ICAM-1 Ab and MDMs with LFA-1 Ab for 24 h. % of syncytia formation in the different groups after overnight ICAM-1 Ab treatment of U1 10 K EV sets (**A**) and U1 100 K EV sets (**B**) and after 3 h incubation of these treated EVs on naïve MDMs. % of syncytia formation in the different groups after overnight MDMs treatment with LFA-1 Ab and on these MDMs 3 h incubations of U1 10 K (**C**), and 100 K (**D**) EV sets. N= 3 (MDMs from three different biological donors); cell: EV cell ratios = 1:10,000; data are shown as mean +/− SEM and represent at least three independent assays done in triplicate. * *p* < 0.05; ** *p* < 0.01; *** *p* < 0.001, **** *p* < 0.0001.

## Data Availability

Not applicable.

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
