# Peer review of "Methamphetamine Induces the Release of Proadhesive Extracellular Vesicles and Promotes Syncytia Formation: A Potential Role in HIV-1 Neuropathogenesis"

_viruses, 2022, doi:10.3390/v14030550_

Round 1

Reviewer 1 Report

The manuscript has merit by addressing the role of extracellular vesicles (EVs) in HIV neuropathogenesis and toxicity, particularly in the presence of Methamphetamine (Meth). The results indicating a role for adhesion molecules in the induction of syncytia are interesting. The design took advantage of a series of in vitro systems using cell lines and primary monocyte-derived macrophages, to mimic infection and signs of pathogenesis. The idea is sound and the design is adequate, but the overall description and interpretation must be revised. This reviewer has identified a number of small inconsistencies, from the abstract to methods, lack of evidence for critical assumptions, and poor discussion about mechanisms operating in the context of ART. The concerns will be numbered below, not as major or minor issues, but as they appear in the text, followed by overall comments and suggestions.

  1. The abstract is not clear about the fact that the majority of the experiments was a actually performed in cell lines, U937 and U1. As written, the work in cell lines is omitted, misleading the reader to understand that EVs were derived from MDMs, or that primary cells were infected. The abstract needs to be clear.
  2. The abstract states that HIV+ Meth users have higher plasma viral load, which may be misleading. In the introduction, references that are cited are about adherence and in the discussion, cited studies performed in monkeys indicate that CSF but not plasma have higher viral load due to METH. These consistency between statements and citations needs to be reexamined.
  3. There are no comments about the latent status of U1 cells, the significance and meaning for the findings.
  4. In methods Meth and ART treatment, only 10uM and 50uM are described, but figures also include 100uM and 250uM, which may affect viability.
  5. What is the rationale for the chosen antiretrovirals in this study?
  6. In Statistics, multiple comparison analysis must be described, since the figures suggest that potshot tests were applied.
  7. In Figure 1, all axis should match.
  8. Is there an explanation for why there is a decrease in cytotoxicity at 6h in U937 cells?
  9. U937 cells do not have a clear dose-dependent toxicity effect, while U1 cells lack a time-dependent toxicity effect. By keeping axis uniform, the effect of HIV in U1 cells may be better visualized.
  10. On page 7, line 262, it says IL1b and TNFa were unregulated by Meth-U1 cells. It does not mention that U937 experienced the same phenomenon.
  11. Figure 2B would benefit from a line on 1-fold, as done for Figure 2A.
  12. Figure 3 may include a graph with normalized band densities. How many times was the experiment performed?
  13. It is important to acknowledge that ART did not decrease Nef or ICAM1. Quite the opposite. That indicates that the effect of ART on syncytia may be independent from the factors evaluated here. A study on EVs load may be more adequate to explain such effect. This must be better developed in the discussion. The experiments shown here DO NOT explain the effect of ART.
  14. More experiments are necessary to prove the role of ICAM and LFA1 in EV fusion.
  15. A better discussion on the significance of EVs of different sizes and implications may be necessary.

Author Response

Our response to reviewer's comments

We thank reviewers for their valuable suggestions. Our responses are italicized and are blue after each query.

Reviewer(s)' Comments to Author:
Reviewer: 1
Comments and Suggestions for Authors.
Comment:
The manuscript has merit by addressing the role of   extracellular vesicles (EVs) in HIV neuropathogenesis and toxicity, particularly in the presence of Methamphetamine (Meth). The results indicating a role for adhesion molecules in the induction of syncytia are interesting. The design took advantage of a series of in vitro systems using cell lines and primary monocyte-derived macrophages, to mimic infection and signs of pathogenesis. The idea is sound and the design is adequate, but the overall description and interpretation must be revised. This reviewer has identified a number of small inconsistencies, from the abstract to methods, lack of evidence
for critical assumptions, and poor discussion about mechanisms operating in the context of ART.
The concerns will be numbered below, not as major or minor issues, but as they appear in the text, followed by overall comments and suggestions.
We are thankful to the reviewer for appreciating our idea, experiments design, and overall study. Following each concern/question, we have revised our manuscript as suggested.
Concerns:
1. The abstract is not clear about the fact that the majority of the experiments was a actually performed in cell lines, U937 and U1. As written, the work in cell lines is omitted, misleading the reader to understand that EVs were derived from MDMs, or that primary cells were infected. The abstract needs to be clear.
We are thankful to the reviewer for pointing this out. We have revised the abstract to alleviate the concern.
2. The abstract states that HIV+ Meth users have higher plasma viral load, which may be misleading. In the introduction, references that are cited are about adherence and in the
discussion, cited studies performed in monkeys indicate that CSF but not plasma have higher viral load due to METH. These consistency between statements and citations needs to be reexamined.
We have changed plasma viral load to viral load in the brain. The introduction and discussion sections have been modified accordingly to address the reviewer's concerns.
3. There are no comments about the latent status of U1 cells, the significance and meaning for the findings.
We have now included this in our discussion. Please see lines 423-435 of the revised manuscript.

4. In methods Meth and ART treatment, only 10uM and 50uM are described, but figures also include 100uM and 250uM, which may affect viability.
We thank the reviewer to point this discrepancy. We have corrected this typo in the methods section.
5. What is the rationale for the chosen antiretrovirals in this study?
We apologize if we haven’t been clear about why we have chosen antiretrovirals in this study. We used the NIH Guidelines for the Use of Antiretroviral Agents in Adults and Adolescents with HIV developed by the DHHS Panel on Antiretroviral Guidelines for Adults and Adolescents- a working group of the office of AIDS research advisory council. (https://clinicalinfo.hiv.gov/sites/default/files/guidelines/documents/AdultandAdolescentGL.pdf [secure-web.cisco.com])
6. In Statistics, multiple comparison analysis must be described, since the figures suggest that potshot tests were applied.
Thank you. We added this analysis as suggested in the statistics section.
7. In Figure 1, all axis should match.
We have changed the axis of figure 1 as advised by the reviewer.
8. Is there an explanation for why there is a decrease in cytotoxicity at 6h in U937 cells?
Although we see some decrease in cytotoxicity at 6h, it is not significant. This could be because the acute dose of Meth can possibly increase a cellular antioxidant defense mechanism and may initially provide some toxic protection.
9. U937 cells do not have a clear dose-dependent toxicity effect, while U1 cells lack a time-dependent toxicity effect. By keeping axis uniform, the effect of HIV in U1 cells may be better
visualized.
We have changed the axis of figure 1 as advised by the reviewer.
10. On page 7, line 262, it says IL1b and TNFa were unregulated by Meth-U1 cells. It does not mention that U937 experienced the same phenomenon.
We have mentioned that IL1b and TNFa were upregulated by Meth in U937 cells on page 7, line 272.
11. Figure 2B would benefit from a line on 1-fold, as done for Figure 2A.
We thank the reviewer for pointing this out. A line on 1-fold has been added in Figure 2B.
12. Figure 3 may include a graph with normalized band densities. How many times was the experiment performed?

We have added proteins bands’ average densitometry value of the in form of a graph in the supplementary figure1 as advised by the reviewer. We performed three independent experiments for all our studies. EVs isolated from these experiments were used for both cell treatments thand the western was performed in duplicates.
13. It is important to acknowledge that ART did not decrease Nef or ICAM1. Quite the opposite.
That indicates that the effect of ART on syncytia may be independent from the factors evaluated here. A study on EVs load may be more adequate to explain such effect. This must be better developed in the discussion. The experiments shown here DO NOT explain the effect of ART.
We agree with the reviewer’s point that ART may have an independent impact on syncytia formation. We have added discussion section to address reviewer's concern in page 13, lines 423-435.
14. More experiments are necessary to prove the role of ICAM and LFA1 in EV fusion.
Antibody blocking of ICAM1 on EV or LFA-1 on MDMs have significantly decreased multinucleated cell formation, indicating their potential role in cell clustering. We believe that further molecular experiments would not add more knowledge to the experiments and out of the scope of the current manuscript.
15. A better discussion on the significance of EVs of different sizes and implications may be necessary.
Thank you for a valuable suggestion. We recently showed that Meth could induce the release of different sizes of EV(Chand et al. 2021; https://doi.org/10.1002/jev2.12177). We have now added these points in the discussion section (lines 470-476) to address reviewer's concern.

Reviewer 2 Report

The manuscript entitled   Role of Methamphetamine Induced Extracellular Vesicles in  HIV Neuropathogenesis and Toxicity.

This study evaluated the effect of methamphetamine on extracellular vesicles production by macrophages cells line. It is relevant in the extracellular vesicle research field.

This manuscript proposes an interesting explanation for the potential role of Methamphetamine on the production of pro adhesive EV. I suggest changing the title to increase adequation with showing results. Methamphetamine induced pro adhesive extracellular vesicles by macrophage cell lines: a potential role in neuropathogenesis

Major corrections:

I suggest rewriting the introduction in link with the result shows.   

Minor questions

Line 98: What rationale to treat the cells with Methamphetamine before ART  ?

What is the rationale to treat the cells with 100 uM of meth for the gene expression since this concentration does not induce changes in the EVs released in U1 cells  ?

Originality of study

This article is original.  

Scientific quality

It is a well-written manuscript, and the figures are adequately presented.

Impact of the research

The impact of this work is excellent.

Author Response

Reviewer: 2
Comments:
The manuscript entitled Role of Methamphetamine Induced Extracellular Vesicles in HIV Neuropathogenesis and Toxicity.
This study evaluated the effect of methamphetamine on extracellular vesicles production by macrophages cells line. It is relevant in the extracellular vesicle research field.
This manuscript proposes an interesting explanation for the potential role of Methamphetamine on the production of pro adhesive EV. I suggest changing the title to increase adequation with showing results. Methamphetamine induced pro adhesive extracellular vesicles by macrophage cell lines: a potential role in neuropathogenesis
We thank the reviewer for appreciating our study. We have changed the title toMethamphetamine induces the release of proadhesive extracellular vesicles and promotes syncytia formation: a potential role in HIV-1 neuropathogenesis

Major corrections:
I suggest rewriting the introduction in link with the result shows.
We thank the reviewer for the advice. We have modified our introduction to link it with our results.
Please see the revised introduction marked in red.
Minor questions
Line 98: What rationale to treat the cells with Methamphetamine before ART?
We thank the reviewer for raising an important point. Individuals on ART usually have a history of drug use; in fact, people abusing Meth are at the highest risk of infection. Therefore, we chose to treat cells first with Meth and then with ART; this regimen will also show us the effect of Meth alone on HIV infection and in the presence of ART.
What is the rationale to treat the cells with 100 uM of meth for the gene expression since this concentration does not induce changes in the EVs released in U1 cells?
We thank the reviewer for asking us to clarification regarding the use of Meth concentration. In our recent publication, we found that when microglial cells were treated with 100 μM of Meth, they showed upregulation of several genes involved in EV biogenesis (Chand et al. 2021; https://doi.org/10.1002/jev2.12177). We followed the same regimen in U937 and U1 cells.

Reviewer 3 Report

In this study the authors show how HIV-1 infection and Meth-treatment in synergy affects EV-biogenesis in monocyte-derived macrophages. The manuscript is well-written. The analysis was done using appropriate statistical method and the conclusion is in line with the experimental findings. This research will help to better understand the pathology of CNS infection by HIV-1. However, there are some concerns that I highly recommend to address:

  1. My first concern refers to the lack of information on using chronically infected HIV-1 promonocytic cells as a model for HIV-1 infection.
  2. The authors must discuss reasons for differences observed in particles released between the 10K and 100K EV (Figure 1B, C).
  3. The authors must discuss why the cells are treated with Meth for 24h in Figure 2 but for 48h in Figure 1 and 3?
  4. Figure 2. The genes expression levels may also significantly differ in HIV-1 infection. It will be interesting to see if target genes differ not only between controls and Meth but also between U937 and U1 cells after Meth treatment.
  5. Figure 2. Only qPCR method was used for the results. Please supplement the protein quantification at least in the two or three most representative genes.
  6. The author should add a positive control in Figure 3 gfp120 and VCAM Western Blot.

Author Response

We thank reviewers for their valuable suggestions. Our responses are italicized and are blue after each query.

All our changes have been marked in red in the revised manuscript.

Reviewer: 3

 Comments:

In this study the authors show how HIV-1 infection and Meth-treatment in synergy affects EV-biogenesis in monocyte-derived macrophages. The manuscript is well-written. The analysis was done using appropriate statistical method and the conclusion is in line with the experimental findings. This research will help to better understand the pathology of CNS infection by HIV-1. However, there are some concerns that I highly recommend to address:

We are thankful to the reviewer for recognizing the importance of our study and the flow of the manuscript. We have revised the manuscript as suggested by the reviewer.

  1. My first concern refers to the lack of information on using chronically infected HIV-1 promonocytic cells as a model for HIV-1 infection.

We thank the reviewer for the advice. We have added information on using chronically infected HIV-1 promonocytic cells as a model for HIV-1 infection in the result sections (page 5, lines 236-239).

  1. The authors must discuss reasons for differences observed in particles released between the 10K and 100K EV (Figure 1B, C).

We thank the reviewer for the suggestion. We included the reason in the discussion (page 12, lines 386-391)

  1. The authors must discuss why the cells are treated with Meth for 24h in Figure 2 but for 48h in Figure 1 and 3?

We wanted to screen for EV biogenesis genes that were activated immediately after a single dose treatment for 24h. Since half-life of Meth is ~10-11h , we chose to treat the cells for 24h with one dose of 100 μM Meth to identify genes that are activated immediately after acute drug exposure. We chose to use 48h for the rest of the studies as that replicates a “binge” scenario in human drug users. These points are made clear in results section page 5, lines 244-247 and page 7, lines 272-273.

  1. Figure 2. The genes expression levels may also significantly differ in HIV-1 infection. It will be interesting to see if target genes differ not only between controls and Meth but also between U937 and U1 cells after Meth treatment.

We thank the reviewer for the suggestion and we agree that there might be changes, however since the analysis would not add any significant knowledge to the present study, we have chosen not to perform the analysis.

  1. Figure 2. Only qPCR method was used for the results. Please supplement the protein quantification at least in the two or three most representative genes.

We were primarily interested in identifying immediate activation of EV biogenesis genes after drug exposure. We used a quantitative Taqman qPCR as opposed to SYBR green which is a highly sensitive RT-PCR method and therefore eliminated the use of Western blot.

  1. The author should add a positive control in Figure 3 gfp120 and VCAM Western Blot.

We used HSC70 as a positive control for EVs. The densitometric values are now added as supplementary figure 1.

Round 2

Reviewer 1 Report

All questions have been addressed.

Reviewer 2 Report

The answers are satisfactory to me.

Reviewer 3 Report

The authors successfully addressed all my concerns and the manuscript is now acceptable for publication at this journal.